# ST-Veto: Spatio-Temporal Token Veto for Diffusion MLLMs via Taylor Prediction and Visual Grounding

Keuntae Kim [1][*]   Beomseok Lee [2][*]   Hyunwoo Kim [3][*]   Yong Suk Choi [1][†]

## Abstract

Vision Language Models (VLMs) achieve strong reasoning with Chain-of-Thought (CoT) prompting but incur high sequential-generation cost, error accumulation, and limited self-correction. Diffusion Multimodal Large Language Models (dMLLMs) unmask tokens in an order-agnostic process, improving efficiency and enabling iterative refinement, yet their reasoning and how to enhance it remain underexplored. We propose a training-free method, Spatio-Temporal Token Veto (ST-Veto), which leverages the ability to observe all token positions at each diffusion step. Rather than relying only on current-step confidence, ST-Veto vetoes temporally unstable tokens via second-order Taylor prediction of confidence dynamics and filters weakly grounded tokens using image-attention mass, swapping them with safer candidates. Across multiple dMLLMs and multimodal reasoning benchmarks, ST-Veto consistently outperforms standard decoding policies and prior VLM reasoning methods, improving accuracy by up to 9% with no additional training or generation cost. Analyses show that ST-Veto steers generation toward higher-confidence, better-grounded paths.

## 1. Introduction

Large Language Models (LLMs) have established a strong foundation in NLP tasks through next-token prediction. Building upon this, Vision Language Models (VLMs), which integrate LLMs as backbones with vision encoders, have achieved strong results in the multimodal domain (Alayrac et al., 2022; Li et al., 2023b;a; Wang et al., 2024; Zhang et al., 2024a). Central to the success of both LLMs and VLMs is the Auto-Regressive (AR) mechanism, which generates tokens sequentially. Furthermore, conditioning on previously generated tokens, such as via Chain-of-Thought (CoT) prompting, has substantially enhanced the reasoning capabilities of AR models (Wei et al., 2022; Zhang et al., 2022; Wang et al., 2022).

However, the sequential nature of AR models incurs high latency and inference costs. A critical limitation is the difficulty of self-correction: because the model continually consumes its own generated tokens as input, errors can easily propagate and amplify hallucinations (Ji et al., 2023; Pan et al., 2023). Recent diffusion-based language models instead generate tokens through order-agnostic denoising and iterative refinement (Li et al., 2022; Gong et al., 2022; Sahoo et al., 2024b; Nie et al., 2025). Diffusion Large Language Models (dLLMs) and diffusion Multimodal Large Language Models (dMLLMs) leverage these properties to improve efficiency and enable bidirectional context modeling, facilitating capabilities such as self-correction (Li et al., 2025; Yang et al., 2025; You et al., 2025).

Despite this potential, d(M)LLMs remain in the nascent stages of research and have not yet reached the maturity of AR models, particularly regarding reasoning. Prior works have primarily emphasized efficiency by restricting evaluation to relatively simple tasks with short generation lengths (e.g., 16 or 32 tokens) (Li et al., 2025; Yang et al., 2025). As a result, it remains unclear whether dMLLMs can sustain non-trivial reasoning under long-generation settings or how they internally perform inference. For instance, whether dM-LLMs can generate coherent and faithful rationales when given sufficient generation length has received limited attention. Moreover, since standard CoT prompting does not directly translate to the non-sequential generation paradigm, a key open question is how to improve reasoning reliability in diffusion models (Sahoo et al., 2025).

In this study, we analyze the reasoning capabilities of dM-LLMs and propose a method to enhance them. We evaluate two state-of-the-art dMLLMs, LaViDa-Reason (Li et al., 2025) and MMaDA-MixCoT (Yang et al., 2025), on benchmarks requiring multimodal understanding and reasoning (Chen et al., 2024; Liu et al., 2024; Lu et al.,

*Equal contribution  [1]ktkpv94@hanyang.ac.kr, Hanyang University;  [2]qpwoeiru6486@gmail.com, LG Electronics; [3]kimhw199807@gmail.com, KT Corporation;. Correspondence to: Yong Suk Choi <cys@hanyang.ac.kr>.

*Proceedings of the 43$^{rd}$ International Conference on Machine Learning*, Seoul, South Korea. PMLR 306, 2026. Copyright 2026 by the author(s).

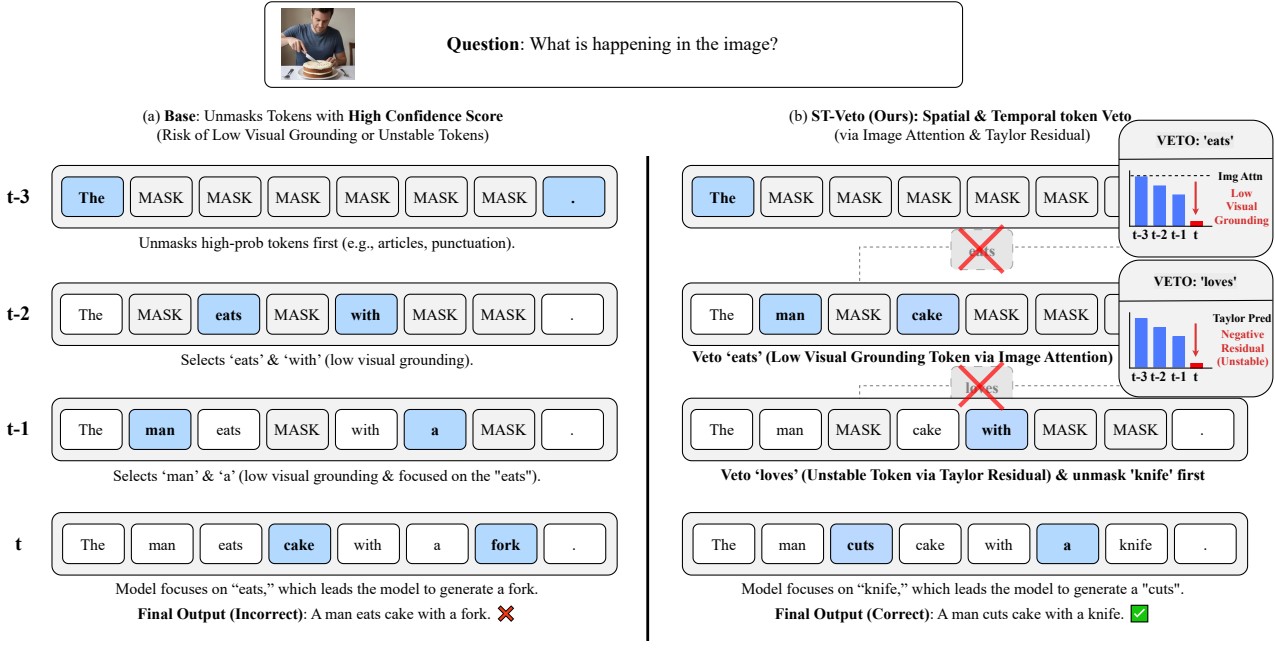

*Figure 1.* Overview of our work. The base decoder unmasks tokens according to current-step confidence, whereas ST-Veto applies temporal-stability and visual-grounding checks before committing tokens to the denoising trajectory.

2022) while allowing sufficiently long generation lengths. Crucially, we observe that directly applying state-of-the-art (SOTA) reasoning methods developed for VLMs is ineffective under the diffusion generation paradigm (Mitra et al., 2024; Zheng et al., 2023; Zhang et al., 2023; Xu et al., 2025). These methods assume an autoregressive chain in which intermediate rationales become stable conditioning context for subsequent tokens. In contrast, dMLLMs repeatedly revise many token positions in parallel, so a token selected at one step can become an unreliable anchor if it is chosen only because it appears confident under the current partially denoised context. These observations motivate a new approach tailored to dMLLMs rather than adapting AR-style reasoning methods.

Our approach leverages a unique characteristic of dMLLMs distinct from AR models: the ability to observe the evolution of all token positions across diffusion time steps. Unlike the generation of AR models, dMLLMs observe all masked positions at every step, unmasking confident tokens while remasking the rest for the subsequent iteration (Sahoo et al., 2024b; Austin et al., 2021). This global visibility suggests that reliable reasoning should not depend only on the instantaneous confidence of a token. Instead, a token should be committed when it is stable across denoising time and meaningfully grounded in the visual input. Capitalizing on this mechanism, we propose **Spatio-Temporal Token Veto (ST-Veto)**, a training-free decoding-time method that exploits spatio-temporal token information to guide generation.

ST-Veto operates via two key mechanisms. First, it tracks token confidence values across diffusion steps and predicts the confidence at the next step using a second-order Taylor expansion. By comparing the actual confidence with the predicted value, ST-Veto identifies unstable tokens whose confidence deviates from the expected trajectory and vetoes them during unmasking selection. Second, it employs image attention mass to veto tokens with low visual grounding, preventing tokens from being unmasked solely due to high textual confidence without sufficient visual support (Favero et al., 2024; Kang et al., 2025; Jiang et al., 2025). Vetoed tokens are replaced only with near-boundary candidates that satisfy both temporal and visual reliability criteria. As a result, ST-Veto encourages the model to build its subsequent context from tokens that are spatio-temporally stable and semantically meaningful.

Across all evaluated models, benchmarks, and settings, ST-Veto consistently outperforms both the baseline decoding strategy and SOTA reasoning methods originally designed for VLMs. Extensive analyses further confirm that ST-Veto functions as intended, guiding dMLLMs toward generation trajectories with higher overall confidence. Notably, ST-Veto requires no additional training and introduces no additional generation cost, while achieving improvements of up to 9%. Our contributions are summarized as follows:

- We systematically evaluate the reasoning behavior of dMLLMs under long-generation settings and show that VLM-based reasoning methods do not transfer effectively to diffusion decoding.

- We propose ST-Veto, a novel training-free decoding-time method that prioritizes spatio-temporally stable tokens by combining confidence-evolution dynamics with visual grounding signals.

**Conflict of Interest Disclosure.** The authors declare no financial conflicts of interest related to this work.

## 2. Background

### 2.1. Diffusion Large Language Models

Diffusion Large Language Models (dLLMs) adapt the diffusion paradigm originally proposed for generating continuous data such as images or audio (Ho et al., 2020; Rombach et al., 2022) to language generation. Early studies applied diffusion to continuous text embeddings (Li et al., 2022; Gong et al., 2022); however, directly modeling discrete token distributions proved challenging, limiting performance. Subsequently, *discrete diffusion* approaches were introduced to learn the probabilistic masking-and-restoration process directly at the token level (Austin et al., 2021; Sahoo et al., 2024a), substantially improving both generation quality and training stability. Recently, dLLMs such as LLaDA (Nie et al., 2025) and Dream (Ye et al., 2025) have emerged, offering a new paradigm that balances inference speed and generation quality through parallel token restoration, effectively addressing the speed–quality trade-off.

Formally, given a discrete token sequence of length $L$, $X_0 = [X_0^1, X_0^2, \ldots, X_0^L]$, we consider a continuous-time masking process parameterized by $t \in [0, 1]$. Let $q(X_t \mid X_s)$ denote the forward noising (masking) transition kernel for $1 \geq t \geq s \geq 0$, which progressively replaces a subset of tokens with a special mask token $[M]$. At $t = 1$, the sequence becomes fully masked, i.e., $X_1 = [M, M, \ldots, M]$. A model $p_\theta$ parameterizes the reverse denoising, learning to reconstruct less-masked states from more-masked ones (e.g., $p_\theta(X_s \mid X_t)$ for $s < t$) (Sahoo et al., 2024a; Nie et al., 2025).

During inference, sampling starts from the fully masked sequence $X_1$ and iteratively transitions to less-masked states by predicting masked tokens using the learned reverse model. Concretely, given the current state $X_t$, the model predicts all masked positions in parallel via $p_\theta(\cdot \mid X_t)$. Then, following a masking schedule, some tokens are *frozen* (kept) while the remaining tokens are *remasked* for further refinement. One simple example is to remask a fraction determined by a schedule $\gamma(t \rightarrow s)$ (e.g., a function of $t$ and $s$) while keeping the others fixed (Sahoo et al., 2024a). Repeating this procedure gradually reduces the masking level and ultimately yields a complete token sequence.

### 2.2. Diffusion Multimodal Large Language Models

Diffusion MLLMs extend discrete diffusion language modeling to conditional generation with multimodal inputs, such as an image $I$ and a prompt $P$. Models like LaViDa (Li et al., 2025) and MMaDA (Yang et al., 2025) learn reverse denoising as a masked-token prediction task conditioned on multimodal signals.

LaViDa encodes an image $I$ into continuous visual features and trains a diffusion language model to predict the original text sequence $X_0$ from a partially masked $X_t$ given $I$ and $P$. The training objective is defined as (Li et al., 2025):

$$\mathcal{L}_{\text{LaViDa}} = -\mathbb{E}_{t,I,P,X_0,X_t}\Big[w(t) \log p_\theta(X_0 \mid I, P, X_t)\Big], \tag{1}$$

where $w(t)$ is a time-dependent reweighting (often chosen as $w(t) = 1/t$). With masked positions $\mathcal{M}_t = \{i \mid X_t^i = [M]\}$, the loss effectively applies only to those positions. To fit the constraints of the model, we reformulate the objective as:

$$\mathcal{L}_{\text{LaViDa}} = -\mathbb{E}_{t,I,P,X_0,X_t}\Bigg[w(t) \sum_{i \in \mathcal{M}_t} \log p_\theta(X_0^i \mid I, P, X_t)\Bigg]. \tag{2}$$

In contrast, MMaDA unifies text and image into a single discrete token sequence (Yang et al., 2025). With discrete image tokens $V_0 = [V_0^1, \ldots, V_0^{L_v}]$ and text tokens $X_0 = [X_0^1, \ldots, X_0^{L_x}]$, the unified representation is:

$$U_0 = [P; X_0; V_0]. \tag{3}$$

The forward process masks a subset of positions to form $U_t$, and the model predicts the masked tokens. The unified pretraining objective is given by:

$$\mathcal{L}_{\text{MMaDA}} = -\mathbb{E}_{t,U_0,U_t}\Bigg[\frac{1}{|\mathcal{M}_t|} \sum_{i \in \mathcal{M}_t} \log p_\theta(U_0^i \mid U_t)\Bigg], \tag{4}$$
$$\text{where } t \sim \mathcal{U}(0, 1).$$

Unlike LaViDa, which uses continuous visual features for conditioning, MMaDA tokenizes images discretely and optimizes a single diffusion objective over both text and image tokens, enabling unified multimodal understanding and text-to-image generation.

### 2.3. Reasoning in dMLLMs

Current research on multimodal reasoning primarily targets autoregressive (AR) models, employing Chain-of-Thought (CoT) strategies that rely on generating intermediate rationales sequentially (Wei et al., 2022; Zhang et al., 2023).

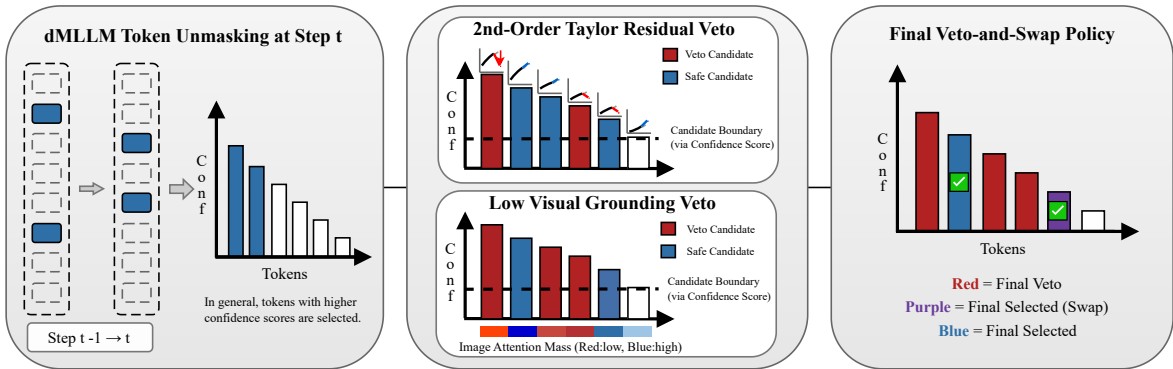

*Figure 2.* Overview of ST-Veto. Tokens are vetoed based on the Taylor residual and low visual grounding, and are replaced by safe tokens that are unmasked instead.

However, these methods are structurally incompatible with the diffusion generation paradigm. Unlike AR models that condition current generation on a fixed history of preceding tokens, dMLLMs update all tokens simultaneously through iterative denoising (Sahoo et al., 2024b). Consequently, the standard "generate-then-condition" mechanism fails in diffusion decoding because there is no stable, causal sequence to anchor intermediate reasoning steps during the bidirectional refinement process. This fundamental mismatch underscores the necessity for diffusion-native reasoning approaches that leverage the global visibility of token evolution, rather than forcing sequential dependencies.

## 3. Method

We operationalize the self-correction capability of diffusion models through a mechanism we term "Veto-and-Swap". A common diffusion decoding policy unmasks the currently most confident tokens at each step. However, an unmasking decision made only from the instantaneous context at step $t$ can be unreliable: a token may be highly ranked because it is locally obvious in the current partially denoised sequence, while its confidence trajectory is unstable or its prediction is weakly supported by the image. ST-Veto therefore seeks to secure tokens that are both temporally stable and spatially grounded before they become fixed context for later denoising steps. Our approach is motivated by three key insights derived from the properties of diffusion models:

- **Stability via Veto-and-Swap:** Directly manipulating logits or attention distributions can destabilize the diffusion process. Instead, a discrete veto-and-swap policy offers a more robust control mechanism.

- **Temporal Stability:** Unlike autoregressive models, dMLLMs provide visibility into all masked token positions at every step $t$, allowing us to leverage the evolution of confidence over time rather than relying only

on the current-step score.

- **Visual Anchoring:** Since dMLLMs generate tokens in parallel, we can prioritize unmasking tokens that are supported by visual evidence, using them as spatial anchors to guide subsequent generation.

**Notation.** Let $\mathbf{c}^{(t)} \in \mathbb{R}^n$ denote the confidence score vector of the currently masked token positions at diffusion step $t$, where $n$ is the number of masked positions. Each element $c_i^{(t)}$ is the model confidence assigned to the predicted token at position $i$. We aim to unmask $k$ positions at this step. Let $\mathcal{I}$ be the set of indices corresponding to image tokens, and let $A^{(\ell)} \in \mathbb{R}^{L_{\text{seq}} \times L_{\text{seq}}}$ denote the head-averaged attention matrix at Transformer layer $\ell$, where $L_{\text{seq}}$ is the total multimodal sequence length. We denote the number of Transformer layers by $N_{\text{lay}}$.

### 3.1. Taylor Residual (Temporal Instability Signal)

In diffusion-based generation, token-level confidence scores are expected to evolve smoothly as the sequence is progressively denoised. However, abrupt fluctuations often signal instability: a token may look confident under the current noisy context but become unsupported once neighboring tokens are refined. Such premature commitments can produce unstable anchors for later steps. To identify these cases, we predict the expected confidence via a second-order Taylor expansion using finite-difference velocity and acceleration computed from a local 4-step history over the same masked positions, $\{\mathbf{c}^{(t-3)}, \dots, \mathbf{c}^{(t)}\}$:

$$
\begin{aligned}
\hat{\mathbf{c}}^{(t)} = \mathbf{c}^{(t-1)} + \big(\mathbf{c}^{(t-1)} - \mathbf{c}^{(t-2)}\big) \\
+ \frac{1}{2}\Big[\big(\mathbf{c}^{(t-1)} - \mathbf{c}^{(t-2)}\big) - \big(\mathbf{c}^{(t-2)} - \mathbf{c}^{(t-3)}\big)\Big].
\end{aligned} \tag{5}
$$

The instability signal (Taylor residual) for token $i$ is then defined as the deviation from this trajectory:

$$e_i = c_i^{(t)} - \hat{c}_i^{(t)}. \tag{6}$$

Here, large negative residuals ($e_i \ll 0$) indicate that the confidence has dropped significantly below expectation, signaling temporal instability.

### 3.2. Image Attention (Grounding Signal)

Temporal stability alone does not ensure that an unmasked token is semantically grounded in the visual input. A token can be stable because it is a frequent or textually plausible continuation, while still being weakly supported by the image. Unlike sequential generation, dMLLMs can commit to token positions in any order, so ST-Veto can prefer tokens that are meaningful in the current multimodal context and visually supported before they influence later denoising. We measure visual grounding by aggregating the attention mass assigned to image tokens $\mathcal{I}$ across a set of intermediate layers:

$$\mathcal{L}_{\text{attn}} = \left\{ \frac{N_{\text{lay}}}{3}, \frac{N_{\text{lay}}}{2}, \frac{2N_{\text{lay}}}{3} \right\}. \tag{7}$$

When these indices are non-integers, we use the nearest valid layer index. For a token $i$ at layer $\ell$, we define the image attention ratio $\rho_i^{(\ell)}$ as:

$$\rho_i^{(\ell)} = \frac{\sum_{j \in \mathcal{I}} A_{i,j}^{(\ell)}}{\sum_{p=1}^{L_{\text{seq}}} A_{i,p}^{(\ell)}}. \tag{8}$$

The final grounding signal $m_i$ is obtained by averaging across the selected layers:

$$m_i = \frac{1}{|\mathcal{L}_{\text{attn}}|} \sum_{\ell \in \mathcal{L}_{\text{attn}}} \rho_i^{(\ell)}. \tag{9}$$

### 3.3. Veto-and-Swap Policy

A standard decoding strategy is to unmask the set of tokens with the highest confidence scores at the current step. This greedy rule is efficient, but it only reflects the current partially denoised context. ST-Veto instead treats the top-$k$ set as a proposal and checks whether each proposed token is spatio-temporally reliable. Let $\mathcal{T}_k$ denote the indices of the top-$k$ tokens:

$$\mathcal{T}_k = \left\{ i \mid \text{rank}_{\downarrow}(c_i^{(t)}) \leq k \right\}, \tag{10}$$

where $\text{rank}_{\downarrow}$ ranks confidence scores in descending order. Let $c_{(k)} = \min_{i \in \mathcal{T}_k} c_i^{(t)}$ represent the confidence threshold. While efficient, this greedy selection often includes unstable or weakly grounded tokens. We therefore apply a conservative policy based on robust statistics.

**Robust statistics.** For a signal $X \in \{e, m\}$, we compute robust location and scale estimates: $\mu_X = \text{median}(X)$ and $\sigma_X = \text{MAD}(X)$.

**Veto and Safe Criteria.** A token $i$ is flagged for veto if either its temporal residual or visual grounding score deviates negatively from the population distribution:

$$\mathcal{C}_{\text{veto}}(i) \iff (e_i < \mu_e - \sigma_e) \lor (m_i < \mu_m - \sigma_m). \tag{11}$$

Conversely, a token is considered safe only if it satisfies both stricter temporal stability and visual grounding requirements:

$$\mathcal{C}_{\text{safe}}(i) \iff (e_i \geq \mu_e + \sigma_e) \land (m_i \geq \mu_m + \sigma_m). \tag{12}$$

**Veto and Swap sets.** We define the veto set $\mathcal{V}$ as the risky tokens currently within the top-$k$ selection:

$$\mathcal{V} = \{i \in \mathcal{T}_k \mid \mathcal{C}_{\text{veto}}(i)\}. \tag{13}$$

To fill the vacated slots, we identify a set of swap candidates $\mathcal{S}$. To prevent destabilization from low-confidence tokens, candidates must be safe and reside within a "candidate line" (a near-boundary region around $c_{(k)}$):

$$\mathcal{S} = \left\{ i \notin \mathcal{T}_k \mid \mathcal{C}_{\text{safe}}(i) \land |c_i^{(t)} - c_{(k)}| \leq \sigma_{\mathbf{c}} \right\}, \tag{14}$$

where $\sigma_{\mathbf{c}} = \text{MAD}(\mathbf{c}^{(t)})$.

**Final Selection.** We replace the tokens in $\mathcal{V}$ with the highest-confidence candidates from $\mathcal{S}$. If $|\mathcal{S}| < |\mathcal{V}|$, we only perform partial swaps. Thus, ST-Veto does not force a replacement when no sufficiently stable and grounded alternative exists. The final mask is updated accordingly for the next step, allowing the model to build subsequent context from tokens that are more reliable across time and more meaningful with respect to the image.

## 4. Experimental Setup

**Benchmarks.** We evaluate our proposed method on four diverse benchmarks: **M3CoT** (Chen et al., 2024), **ScienceQA** (Lu et al., 2022), **MMBench** (Liu et al., 2024), and **V\*Bench** (Wu & Xie, 2023). **M3CoT** is designed to assess multimodal chain-of-thought (CoT) reasoning across broad domains, including science, mathematics, and commonsense. **ScienceQA** evaluates knowledge-based multimodal reasoning by integrating scientific concepts with visual information. **MMBench** comprises diverse evaluation splits to measure general visual perception and visual reasoning capabilities. **V\*Bench** focuses on high-resolution visual question answering (VQA), emphasizing fine-grained visual understanding. We follow the official evaluation protocols and report results on the standard splits for all benchmarks.

*Table 1.* Main results table. X/Y denotes the generation length $L$ and the number of steps $T$. Bold indicates the best performance. All experiments were conducted three times with different random seeds, and the reported results are averaged.

| Model | Method | M$^3$CoT | | MMBench | | SQA-IMG | | V* Bench | |
|---|---|---|---|---|---|---|---|---|---|
| | | 128/64 | 256/128 | 128/64 | 256/128 | 128/64 | 256/128 | 128/64 | 256/128 |
| *LaViDa* | Base-Confidence | 27.96 | 29.55 | 33.96 | 31.55 | 45.81 | 40.06 | 35.60 | 37.17 |
| | Base-Margin | 27.88 | 28.22 | 32.99 | 31.68 | 46.01 | 38.99 | 35.08 | 37.70 |
| | Base-Entropy | 27.91 | 29.23 | 32.67 | 31.92 | 45.89 | 39.17 | 34.55 | 36.65 |
| | DDCoT | 26.99 | 29.11 | 33.02 | 30.98 | 45.99 | 38.81 | 35.08 | 35.60 |
| | CCoT | 26.57 | 28.98 | 33.19 | 31.04 | 45.21 | 38.90 | 36.65 | 36.65 |
| | SCAFFOLD | 26.61 | 28.81 | 33.24 | 31.22 | 44.98 | 39.25 | 35.60 | 35.08 |
| | ICoT | 27.82 | 29.29 | 33.22 | 31.43 | 45.68 | 39.17 | 37.17 | 37.17 |
| | **ST-Veto (Ours)** | **30.21** | **31.87** | **38.05** | **38.25** | **47.80** | **48.54** | **42.94** | **41.36** |
| *MMaDA* | Base-Confidence | 26.32 | 25.11 | 33.75 | 32.04 | 44.72 | 41.70 | 34.03 | 31.41 |
| | Base-Margin | 26.22 | 25.54 | 33.22 | 31.25 | 44.99 | 41.55 | 32.98 | 32.46 |
| | Base-Entropy | 26.36 | 25.32 | 33.53 | 31.87 | 44.23 | 41.98 | 33.51 | 31.41 |
| | DDCoT | 25.44 | 24.45 | 33.21 | 30.04 | 43.99 | 39.98 | 33.51 | 29.84 |
| | CCoT | 26.10 | 24.88 | 34.01 | 31.55 | 44.81 | 40.12 | 34.55 | 29.32 |
| | SCAFFOLD | 25.88 | 24.91 | 33.88 | 30.98 | 44.92 | 40.33 | 31.41 | 28.80 |
| | ICoT | 26.01 | 24.78 | 33.93 | 32.44 | 44.52 | 40.47 | 34.55 | 31.41 |
| | **ST-Veto (Ours)** | **27.52** | **27.74** | **36.55** | **36.74** | **47.25** | **45.31** | **41.36** | **37.17** |

*Table 2.* Time cost table. The unit is inference per second, and the values are averaged over runs on MMbench.

| Method | MMaDA | | LaViDa | |
|---|---|---|---|---|
| | 128/64 | 256/128 | 128/64 | 256/128 |
| Base-Confidence | 5.68 | 11.32 | 4.87 | 12.88 |
| Base-Margin | 5.65 | 11.28 | 4.85 | 12.84 |
| Base-Entropy | 5.68 | 11.33 | 4.88 | 12.89 |
| DDCoT | 10.87 | 19.75 | 9.89 | 22.95 |
| CCoT | 10.93 | 19.48 | 9.92 | 22.94 |
| SCAFFOLD | 10.99 | 19.43 | 9.85 | 22.91 |
| ICoT | 10.96 | 19.89 | 9.87 | 22.87 |
| ST-Veto | 5.73 | 11.39 | 4.89 | 12.92 |

**Models and Baselines.** Our experiments are conducted on two diffusion-based multimodal large language models (dMLLMs) exhibiting strong reasoning capabilities: **LaViDa-llada-reason** (Li et al., 2025) and **MMaDA-8B-MixCoT** (Yang et al., 2025). Both models are fine-tuned to generate rationales and produce progressive outputs via an unmasking-based generation process. For rigorous comparisons, we establish two groups of baselines. First, we compare decoding-policy baselines using three remasking strategies: **Low-confidence (Low-conf)**, **Entropy**, and **Margin** (Sahoo et al., 2024a). Second, we include CoT-style prompting baselines that have demonstrated strong performance in autoregressive VLMs: **CCoT** (Mitra et al., 2024) and **DDCoT** (Zheng et al., 2023), as well as more recent approaches that explicitly enhance vision–language coordination, including **ICoT** (Gao et al., 2025) and **SCAF-FOLD** (Lei et al., 2024). For all prompting baselines, we apply their original prompting templates (and visual prompt-

ing schemes when applicable) while employing Low-conf remasking as the decoding policy for fair comparison.

**Implementation Details.** We evaluate our method by analyzing the speed–quality trade-off—a key advantage of dMLLMs—across varying target output lengths and denoising timesteps. Specifically, we fix the ratio $L/T$ to a standard setting of $0.5$, where $L$ denotes the target generation length and $T$ represents the number of denoising iterations. To ensure strict reproducibility, we perform deterministic decoding by selecting the argmax token at each denoising step (i.e., no temperature scaling). Unless otherwise specified, Low-conf (Sahoo et al., 2024a) serves as the default remasking policy. Additional prompt configurations and detailed experimental settings are provided in Appendix A.1.

## 5. Results

### 5.1. Main Results

Table 1 presents the main evaluation results across four benchmarks. We report the average performance over three random seeds. Experimental settings are denoted as $L/T$, where $L$ represents the target generation length and $T$ indicates the number of denoising steps. Unless otherwise specified, "think mode" refers to enabling rationale generation during decoding, with the maximum number of new tokens capped by $L$.

**Performance of Baselines.** As detailed in Table 1, both LaViDa and MMaDA struggle to attain competitive performance when "think mode" is enabled with $L \in \{128, 256\}$. Notably, they exhibit limited performance even on MM-

*Table 3.* Ablation study table. X/Y denotes the generation length $L$ and the number of steps $T$. Bold indicates the best performance. *w/o Taylor* and *w/o Img* apply the same Veto-and-Swap policy.

| Model | Setting | M³CoT | | MMBench | | SQA-IMG | | V* Bench | |
|---|---|---|---|---|---|---|---|---|---|
| | | 128/64 | 256/128 | 128/64 | 256/128 | 128/64 | 256/128 | 128/64 | 256/128 |
| *LaViDa* | Base | 27.96 | 29.55 | 33.96 | 31.55 | 45.81 | 40.06 | 35.60 | 37.17 |
| | w/o Taylor | 28.77 | 29.64 | 36.04 | 35.94 | 45.86 | 46.52 | 38.22 | 39.27 |
| | w/o Img | 29.68 | 29.11 | 36.22 | 35.22 | 47.47 | 47.89 | 41.36 | 38.22 |
| | **ST-Veto** | **30.21** | **31.87** | **38.05** | **38.25** | **47.80** | **48.54** | **42.94** | **41.36** |
| *MMaDA* | Base | 26.32 | 25.11 | 33.75 | 32.04 | 44.72 | 41.70 | 34.03 | 31.41 |
| | w/o Taylor | 26.66 | 25.99 | 35.12 | 35.52 | 47.15 | 43.99 | 39.79 | 32.84 |
| | w/o Img | 26.88 | 26.24 | 34.98 | 35.10 | 45.56 | 43.87 | 37.17 | 33.51 |
| | **ST-Veto** | **27.52** | **27.74** | **35.55** | **35.74** | **47.25** | **45.31** | **41.36** | **37.17** |

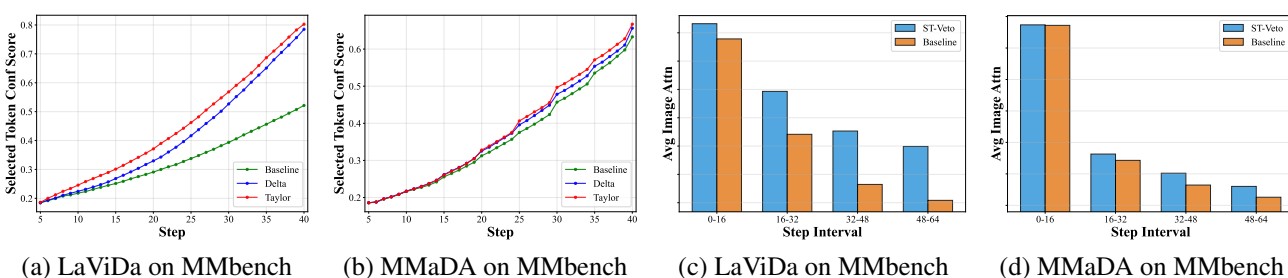

| (a) LaViDa on MMbench | (b) MMaDA on MMbench | (c) LaViDa on MMbench | (d) MMaDA on MMbench |
|---|---|---|---|

*Figure 3.* (a) and (b) show the confidence scores of tokens selected at each step for different methods. (c) and (d) present the relative values of the Average Image Attention.

*Table 4.* Sensitivity to history length on LaViDa. $H = 4$ provides the best cost-performance trade-off; $H = 5$ occasionally matches or exceeds it but requires a higher-order finite-difference estimate.

| Dataset | $L/T$ | Base | Delta | 1st | 2nd | 3rd |
|---|---|---|---|---|---|---|
| MMBench | 128/64 | 33.96 | 36.98 | 37.38 | 38.05 | **38.11** |
| | 256/128 | 31.55 | 36.89 | 37.98 | **38.25** | 38.12 |
| SQA-IMG | 128/64 | 45.81 | 46.69 | 47.25 | 47.80 | **47.91** |
| | 256/128 | 40.06 | 46.36 | 47.60 | **48.54** | 47.55 |
| M³CoT | 128/64 | 27.96 | 28.69 | 29.21 | **30.21** | 30.18 |
| | 256/128 | 29.55 | 30.03 | 30.50 | **31.87** | 31.55 |
| V* Bench | 128/64 | 35.60 | 37.17 | 41.36 | **42.94** | 39.27 |
| | 256/128 | 37.17 | 39.27 | 39.27 | **41.36** | **41.36** |

Bench, a widely adopted benchmark for general VLM evaluation. This underscores a critical bottleneck in current dMLLMs: despite instruction tuning for reasoning (e.g., LaViDa-reason and MMaDA-MixCoT), these models often fail to maintain coherent multi-step rationales under the parallel decoding paradigm. Furthermore, standard remasking strategies (Low-conf, Entropy, Margin) show only marginal performance variance in Table 1, indicating that simple uncertainty-based heuristics are insufficient for complex multimodal reasoning.

**Failure of AR-based Methods.** Methods originally designed for autoregressive (AR) VLMs—including DDCoT, CCoT, and ICoT—prove ineffective when directly applied

to dMLLMs (Table 1). These approaches rely on sequential dependency, generating intermediate contexts (e.g., captions, scene graphs, or rationales) before generating the final answer. However, dMLLMs generate tokens in parallel via iterative refinement, which does not naturally support reusing intermediate tokens as fixed conditioning signals. Consequently, the "generate-then-condition" paradigm of AR methods does not transfer well to diffusion decoding.

**Superiority of ST-Veto.** Across all models, benchmarks, and $L/T$ settings, ST-Veto consistently achieves the best performance (Table 1). Unlike baselines hampered by such structural mismatches, ST-Veto delivers substantial and consistent gains without requiring additional training, multistage generation, or direct modification of attention maps or logits. These results support our hypothesis that effective reasoning control in dMLLMs necessitates mechanisms tailored to the diffusion process rather than direct transplantation of AR-based techniques.

### 5.2. Efficiency

Table 2 compares the inference throughput (samples per second) of different methods on MMBench. All methods are evaluated on identical hardware with the same caching mechanisms in a training-free setting.

**Computational Overhead.** ST-Veto introduces negligible overhead compared to the base model. The computations

*Table 5.* Layer ablation for the image-attention grounding signal on LaViDa. Combining intermediate depths is more robust than using a single layer.

| Dataset | $L/T$ | Base | $L/3$ | $L/2$ | $2L/3$ | All |
|---|---|---|---|---|---|---|
| MMBench | 128/64 | 33.96 | 35.83 | 37.12 | 37.68 | **38.05** |
| | 256/128 | 31.55 | 34.21 | 36.08 | 35.74 | **38.25** |
| SQA-IMG | 128/64 | 45.81 | 46.95 | 47.40 | 46.85 | **47.80** |
| | 256/128 | 40.06 | 44.52 | 46.71 | 47.85 | **48.54** |
| $M^3$CoT | 128/64 | 27.96 | 29.12 | 29.85 | **30.37** | 30.21 |
| | 256/128 | 29.55 | 30.67 | 31.23 | 30.85 | **31.87** |
| V* Bench | 128/64 | 35.60 | 38.74 | 40.31 | 41.36 | **42.94** |
| | 256/128 | 37.17 | 39.27 | 40.31 | **41.88** | 41.36 |

*Table 6.* Sensitivity to the robust threshold on LaViDa. The default MAD cutoff is consistently strong across benchmarks and generation settings.

| Dataset | $L/T$ | Base | 0.5MAD | MAD | 1.5MAD |
|---|---|---|---|---|---|
| MMBench | 128/64 | 33.96 | 36.62 | **38.05** | 37.24 |
| | 256/128 | 31.55 | 36.45 | **38.25** | 36.12 |
| SQA-IMG | 128/64 | 45.81 | 47.12 | **47.80** | **47.80** |
| | 256/128 | 40.06 | 47.45 | **48.54** | 47.22 |
| $M^3$CoT | 128/64 | 27.96 | 29.96 | **30.21** | 28.88 |
| | 256/128 | 29.55 | 31.22 | **31.87** | 31.22 |
| V* Bench | 128/64 | 35.60 | 39.27 | **42.94** | 41.36 |
| | 256/128 | 37.17 | 40.61 | **41.36** | 38.22 |

for second-order Taylor scoring and visual grounding are lightweight and seamlessly integrated into the decoding loop. **Comparison with AR Methods.** In contrast, adapting AR-based methods to dMLLMs typically incurs significant computational costs. To emulate their intended pipeline (context $\rightarrow$ answer), we perform two separate inference calls: one to generate the intermediate context and another to generate the final response conditioned on that context. Single-pass attempts cannot reliably reproduce the intended behavior due to the non-sequential nature of diffusion decoding. As a result, AR-based baselines effectively double the generation cost, whereas ST-Veto operates efficiently within a single diffusion inference process.

### 5.3. Ablation Study

To quantify the contribution of each component, we conduct an ablation study on the Second-Order Taylor Veto and the Visual Grounding Veto (Table 3). Here, w/o Taylor disables the Taylor Veto (using only Visual Grounding), and w/o Img disables the Visual Grounding Veto (using only Taylor); both variants retain the Veto-and-Swap policy.

We observe that employing either component individually yields improvements over the base model on most benchmarks, indicating that both Hessian-based confidence (Taylor) and visual consistency (Visual Grounding) are beneficial for dMLLM reasoning. Crucially, the full ST-Veto achieves the highest performance, demonstrating complementary effects from enforcing both textual logicality and visual grounding. The modular design further facilitates plug-and-play integration with other decoding strategies.

Table 4 further examines the temporal order used by the Taylor-based veto on LaViDa. All history-aware variants outperform the baseline, indicating that temporal confidence dynamics are broadly useful for dMLLM decoding. Among them, the proposed second-order setting ($H = 4$) achieves the strongest overall trade-off: it is best in most settings and avoids the additional higher-order finite-difference estimate required by $H = 5$.

Table 5 analyzes which layers should be used for the visual grounding signal. Single-layer variants already improve over the baseline in most settings, but no individual depth is uniformly optimal across benchmarks. Averaging the intermediate layers ($L/3$, $L/2$, and $2L/3$) gives the most reliable overall behavior, suggesting that different depths provide complementary visual evidence for selecting spatially meaningful tokens.

Table 6 evaluates the robustness of the MAD-based thresholds used in the veto and safe criteria. The relaxed 0.5MAD setting allows more tokens to enter the veto-and-swap procedure, while 1.5MAD is more conservative and can prevent useful swaps. The default MAD cutoff achieves the strongest or tied-strongest performance in nearly all settings, supporting a distribution-adaptive threshold that balances correction and stability.

## 6. Analysis

### 6.1. Methodological Analysis

Table 7 reports, for each model and benchmark, the proportions of tokens in the top-$k$ set that are actually selected through the veto-and-swap procedure, as well as the veto and safe ratios among candidate tokens that meet a certain confidence threshold. Specifically, when unmasking $k$ tokens at each step, the top-$k$ column shows the fraction of tokens that are vetoed and subsequently replaced by candidate tokens. Both LaViDa and MMaDA exhibit similar trends across all benchmarks. Although the veto-and-swap criteria can be adjusted to increase the number of swapped tokens, excessively overriding the model's inherent confidence estimates is undesirable, as assumed in our method design.

The veto ratio among candidate tokens is substantially higher than that in the top-$k$ set. Our analysis reveals a strong correlation between the proposed stability metrics and model confidence: tokens flagged by Taylor residuals or low image attention consistently exhibit lower confidence

*Table 7.* Analysis Table. Metrics are reported for *top-k* (Veto & Swap) and *candidate* (Veto / Safe; in percent with *Veto+Safe*=100) under each configuration (128/64 and 256/128). Candidate ratios for each model/setting are adjusted to be close to the *V\* Bench* ratio.

| Model | Benchmark | 128/64 | | | 256/128 | | |
| | | top-k | candidate | | top-k | candidate | |
| | | Veto & Swap | Veto | Safe | Veto & Swap | Veto | Safe |
|---|---|---|---|---|---|---|---|
| *LaViDa* | M$^3$CoT | 4.8±1.6 | 18.2±5.2 | 81.8±5.2 | 2.6±0.9 | 19.3±3.9 | 80.7±3.9 |
| | MMBench | 5.3±2.2 | 19.0±6.4 | 81.0±6.4 | 3.1±1.3 | 20.1±3.2 | 79.9±3.2 |
| | SQA-IMG | 5.1±2.1 | 18.7±5.6 | 81.3±5.6 | 2.9±1.2 | 19.8±3.1 | 80.2±3.1 |
| | V* Bench | 4.8±6.1 | 18.4±5.9 | 81.6±5.9 | 2.2±3.0 | 19.5±2.9 | 80.5±2.9 |
| *MMaDA* | M$^3$CoT | 5.8±1.3 | 14.5±3.0 | 85.5±3.0 | 7.4±1.8 | 11.8±3.5 | 88.2±3.5 |
| | MMBench | 5.2±1.0 | 14.2±2.4 | 85.8±2.4 | 6.9±1.5 | 11.5±2.7 | 88.5±2.7 |
| | SQA-IMG | 3.9±0.8 | 13.6±2.3 | 86.4±2.3 | 5.5±1.2 | 10.9±3.1 | 89.1±3.1 |
| | V* Bench | 6.1±2.9 | 14.7±2.0 | 85.3±2.0 | 6.3±3.7 | 11.2±2.8 | 88.8±2.8 |

scores. Consequently, the candidate set naturally yields a higher veto ratio, as it contains tokens with lower confidence scores than those in the top-$k$ set. When comparing LaViDa and MMaDA, MMaDA shows a higher proportion of safe tokens. This aligns with the larger performance gains observed for LaViDa relative to MMaDA in Figure 3 and is likely attributable to differences in their training architectures, particularly in how visual information is processed.

## 6.2. Temporal Dynamics Analysis

The core idea of our method is that token unmasking decisions at step $t$ should not be made solely based on the instantaneous state at that step, but should instead account for how token confidence evolves over time as diffusion progresses. To demonstrate the effectiveness of this insight, Figure 3 compares the confidence scores of tokens selected at each step by methods that incorporate temporal information with those that do not.

The results show that methods using Taylor-based veto not only outperform the baseline, but even a simple *delta veto*, which considers only the confidence difference between the current and the immediately preceding step, achieves higher confidence scores than the baseline. The delta signal serves as a first-order alternative to the Taylor residual. These findings indicate that vetoing temporally unstable tokens and swapping them with more stable ones guides the generation toward output paths with higher overall confidence. This trend is consistently observed in all models, with a stronger effect in LaViDa. Moreover, a veto-and-swap policy based solely on confidence at step $t - 1$ is less effective than one that leverages second-order Taylor predictions, highlighting the benefit of modeling higher-order temporal dynamics.

## 6.3. Spatial Dynamics Analysis

Figure 3 presents the results of measuring the Average Image Attention Ratio for the baseline and ST-Veto. Since outliers may appear at individual steps, we visualize the

results by grouping steps into intervals to better capture the overall trend. The results show that ST-Veto consistently achieves higher image attention ratios than the baseline across all steps. This can be interpreted as the effect of vetoing certain tokens that, despite having high confidence scores, exhibit weak visual grounding.

As the model progressively generates and conditions on more text tokens, the overall average image attention naturally decreases as diffusion steps accumulate. In contrast, the gap between the baseline and ST-Veto widens over time, a pattern that closely mirrors the confidence score dynamics observed in the temporal analysis. These findings suggest that the veto-and-swap policy guides the model toward generation paths that are superior not only in terms of confidence scores but also in visual grounding.

## 7. Conclusion

In this work, we investigated how to improve the reasoning capabilities of dMLLMs. Through extensive empirical analysis, we showed that reasoning methods designed for autoregressive models—especially those relying on sequential token dependencies—do not transfer well to diffusion decoding due to fundamental differences in the generation process. To address this gap, we proposed ST-Veto, a training-free decoding-time strategy tailored to dMLLMs. ST-Veto leverages the evolution of token confidence across denoising steps and visual grounding signals to veto unstable or weakly grounded tokens during unmasking, thereby prioritizing more stable and visually supported generations. Experiments across models and benchmarks demonstrate consistent improvements over standard diffusion decoding and adapted VLM reasoning baselines, achieving gains of up to 9% without additional model training or extra denoising steps. We believe these findings establish a solid foundation for diffusion-native reasoning and encourage future research toward more reliable multimodal inference.

## Impact Statement

This paper proposes a training-free inference-time method to improve reasoning in diffusion-based multimodal large language models (dMLLMs) by refining token unmasking decisions. Our contribution is methodological: we introduce a veto-and-swap policy that uses temporal confidence dynamics and image-attention–based grounding signals to reject unreliable tokens and replace them with safer candidates, improving inference stability without updating model parameters.

The method can enable more accurate and better-grounded multimodal outputs without additional training or generation cost. However, it does not eliminate common risks of multimodal generative systems, including incorrect or biased outputs and potential misuse. Because the approach relies on internal signals (confidence and attention) that are imperfect proxies for correctness, responsible deployment should include standard evaluation and safeguards, with human oversight in high-stakes settings.

## Acknowledgements

This work was supported by the Institute of Information and Communications Technology Planning and Evaluation (IITP) grants (No. RS-2025-25422680 and No. RS-2020-II201373), and by the National Research Foundation of Korea (NRF) grant (No. RS-2025-00520618), funded by the Korean Government (MSIT).

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

*Table 8.* Instance-level flip analysis for LaViDa at $L/T = 256/128$. W→R and R→W denote examples that change from wrong to right and from right to wrong, respectively.

| Benchmark | Base | ST-Veto | Δ | W→R | R→W | Flip ratio / $Z$ |
|---|---|---|---|---|---|---|
| MMBench | 31.55 | 38.25 | +6.70 | 518 | 228 | 2.27 / 10.62 |
| SQA-IMG | 40.06 | 48.53 | +8.47 | 410 | 239 | 1.72 / 6.71 |

*Table 9.* Stagewise distribution of Veto-and-Swap events on LaViDa. Values are percentages over all swap events.

| Normalized timestep range | 64 steps | 128 steps |
|---|---|---|
| 0–17% | 18.3 | 19.6 |
| 17–33% | 38.3 | 37.4 |
| 33–50% | 33.7 | 33.4 |
| 50–67% | 8.6 | 6.9 |
| 67–83% | 5.4 | 3.5 |
| 83–100% | 1.8 | 1.0 |

# A. Additional Experiments and Analyses

This appendix provides additional evidence discussed during the review process. We focus on the robustness of ST-Veto to generation length, the role of early denoising steps, comparisons with additional benchmarks, and diagnostic analyses showing that the gains are not artifacts of output format or random decoding perturbations.

## A.1. Prompt Configurations and Experimental Settings

For all benchmarks, we follow the official question format and answer extraction protocol whenever available. The base dMLLMs are evaluated in their reasoning mode, where the model is allowed to generate an intermediate rationale before producing the final answer. For CoT-style baselines, we use the original prompt templates proposed by each method and keep the same diffusion remasking policy as the base decoder unless the method explicitly requires an additional visual prompting component. This keeps the comparison focused on the reasoning or decoding strategy rather than on prompt-format differences.

All main experiments use deterministic decoding with argmax token selection at each denoising step, without temperature sampling. Unless otherwise specified, Low-confidence remasking is used as the default remasking policy. We evaluate the two generation settings used throughout the paper, $L/T = 128/64$ and $L/T = 256/128$, where $L$ is the maximum generation length and $T$ is the number of denoising steps. Additional long-generation results with $L/T = 512/256$ are reported in Appendix A.4.

## A.2. Evaluation Protocol and Output Validity

All reported benchmarks use rule-based or official answer extraction rather than LLM/VLM-as-judge evaluation. Thus, improvements are not caused by stylistic differences in generated rationales. We further verified that both the base decoder and ST-Veto produce zero invalid or unparseable outputs on MMBench and ScienceQA in the settings analyzed below. The gains therefore come from changes in the selected answer option, not from cleaner formatting.

Table 8 reports a paired instance-level flip analysis. If ST-Veto merely introduced random perturbations, the numbers of W→R and R→W flips would be approximately balanced. Instead, the flip ratios are strongly asymmetric, with large positive $Z$-scores. We also observe broad category-level consistency: on MMBench, 17 of 20 categories improve; on ScienceQA, 8 of 9 grade levels and all 3 subjects improve. This pattern supports the interpretation that ST-Veto improves token-level selection rather than relying on stochastic answer changes.

## A.3. Stagewise Veto-and-Swap Dynamics

Table 9 reports where swaps occur during denoising. The swap ratio is larger in early and middle stages, where token commitments can act as anchors for later refinement. ST-Veto is still useful in late stages, but its largest effect comes from avoiding premature commitment to unstable or weakly grounded tokens before they influence subsequent steps.

Table 10. Longer generation results on LaViDa with $L/T = 512/256$.

| Dataset | Base | ST-Veto |
|---|---|---|
| MMBench | 32.54 | **38.12** |
| SQA-IMG | 41.22 | **48.72** |
| M$^3$CoT | 30.58 | **31.76** |
| V* Bench | 36.65 | **42.20** |

Table 11. Additional benchmark results on LaViDa with $L/T = 128/64$. MME subtasks report the official score; POPE and MMMU report accuracy.

| Dataset | Base | CCoT | DDCoT | ICoT | ST-Veto |
|---|---|---|---|---|---|
| MME Exist. | 183.33 | 185.00 | 176.67 | 181.67 | **187.00** |
| MME Count | 133.33 | 126.67 | 137.22 | 136.33 | **143.33** |
| MME Pos. | 86.67 | 90.55 | 81.67 | 89.33 | **91.67** |
| MME Color | 141.67 | 148.33 | 149.17 | 147.67 | **150.00** |
| MME Total | 545.00 | 550.55 | 544.73 | 545.00 | **570.00** |
| POPE | 84.45 | 84.95 | 85.10 | 84.95 | **87.65** |
| MMMU | 30.44 | 31.13 | 30.54 | 31.22 | **33.41** |

## A.4. Generation Length

We additionally test a longer generation setting with $L = 512$ and $T = 256$. ST-Veto maintains its advantage, indicating that the policy does not require a larger rejection ratio as output length grows.

## A.5. Additional Benchmarks

We also evaluate LaViDa at $L/T = 128/64$ on MME (Fu et al., 2023), POPE (Li et al., 2023c), and MMMU (Yue et al., 2024). These results use the same reasoning mode as the main experiments; therefore, they are not directly comparable to reports that restrict generation to only a few final-answer tokens.

## A.6. Training-Free Decoding Baselines

Most existing hallucination-mitigation decoders are designed for autoregressive MLLMs, so they are not protocol-aligned with diffusion decoding. We nevertheless adapt representative training-free baselines, including EAH (Zhang et al., 2024b), OPERA (Huang et al., 2024), and FarSight (Tang et al., 2025). EAH provides the strongest non-ST-Veto gain, suggesting that encouraging image attention is also helpful for dMLLMs. OPERA and FarSight are less compatible because they rely on autoregressive beam search behavior or causal-mask modification.

## A.7. What Tokens Are Vetoed and Swapped?

To better understand the swap behavior, we analyze the part-of-speech distribution of vetoed top-$k$ tokens and accepted swap tokens. Vetoed tokens are dominated by function words, while swapped tokens are more often content words. This is consistent with the early-anchor interpretation: ST-Veto tends to delay unstable high-probability fillers and instead commits to semantically informative tokens when they are temporally stable and visually grounded.

## A.8. Fine-Grained Category Analysis

Table 14 reports representative fine-grained MMBench categories. ST-Veto is especially helpful for structured image-text understanding, physical relations, attribute comparison, and recognition tasks. The small number of negative categories indicates that the policy has limited side effects because it does not directly rewrite logits or attention maps.

*Table 12.* Comparison with additional training-free decoding baselines on LaViDa.

| Dataset | $L/T$ | Base | EAH | OPERA | FarSight | ST-Veto |
|---------|-------|------|-----|-------|----------|---------|
| MMBench | 128/64 | 33.96 | 34.96 | 34.03 | 34.23 | **38.05** |
|         | 256/128 | 31.55 | 33.89 | 31.92 | 31.67 | **38.25** |
| SQA-IMG | 128/64 | 45.81 | 45.52 | 45.75 | 45.22 | **47.80** |
|         | 256/128 | 40.06 | 43.44 | 41.12 | 40.85 | **48.54** |
| $M^3CoT$ | 128/64 | 27.96 | 27.87 | 28.11 | 27.88 | **30.21** |
|         | 256/128 | 29.55 | 29.45 | 29.56 | 29.65 | **31.87** |
| V* Bench | 128/64 | 35.60 | 37.17 | 36.65 | 35.60 | **42.94** |
|         | 256/128 | 37.17 | 39.27 | 35.60 | 36.65 | **41.36** |

*Table 13.* Part-of-speech summary for vetoed and swapped tokens. Percentages report the most frequent groups; remaining categories are grouped as Other.

| Vetoed tokens | | | Swapped tokens | | |
|---------------|-------|----------|----------------|-------|----------|
| Part of speech | Ratio | Examples | Part of speech | Ratio | Examples |
| Determiner | 28.4 | the, a, an | Noun | 29.7 | dog, table, color |
| Preposition | 19.1 | of, in, at, to | Verb | 18.4 | holding, placed, sitting |
| Conjunction | 14.6 | and, or, but | Preposition | 14.2 | in, on, between |
| Other | 37.9 | – | Other | 37.7 | – |

## A.9. Limitations and Future Work

ST-Veto relies on internal confidence and attention signals, which are useful but imperfect proxies. Non-smooth confidence trajectories can make Taylor-based prediction less reliable, and image attention may be diffuse or semantically misaligned with the visual evidence needed for an answer. Our conservative Veto-and-Swap policy mitigates these issues by requiring a safe near-boundary candidate before replacing a token, but it cannot guarantee correction in every case.

The present study focuses on single-image multimodal reasoning with current dMLLMs. Extending the method to multi-image inputs, video understanding, and ultra-long generations will require additional design, particularly for normalizing visual grounding signals across many visual tokens and for handling attention dilution as the generated context grows. We view these settings as important directions for diffusion-native multimodal reasoning.

*Table 14.* Representative fine-grained MMBench category results.

| Category | $N$ | Base | ST-Veto | Diff. |
|---|---|---|---|---|
| Structuralized image-text understanding | 282 | 36.88 | **57.09** | +20.21 |
| Physical relation | 94 | 34.04 | **48.94** | +14.89 |
| Image topic | 140 | 31.43 | **44.29** | +12.86 |
| Action recognition | 215 | 28.37 | **40.47** | +12.09 |
| Attribute comparison | 141 | 32.62 | **44.68** | +12.06 |
| Celebrity recognition | 396 | 25.51 | **27.02** | +1.52 |
| Spatial relationship | 177 | 28.25 | **28.81** | +0.56 |
| Attribute recognition | 264 | **35.61** | 35.23 | -0.38 |
| Image emotion | 200 | **40.50** | 40.00 | -0.50 |
| Identity reasoning | 176 | **26.14** | 25.00 | -1.14 |

