# OpenReview forum: "ST-Veto: Spatio-Temporal Token Veto for Diffusion MLLMs via Taylor Prediction and Visual Grounding"
_ICML.cc/2026/Conference — ICML 2026 regular_

### Official Review · Reviewer_tKTA · 2026-02-20

**Soundness:** 3
**Presentation:** 3
**Significance:** 3
**Originality:** 3
**Overall Recommendation:** 5
**Confidence:** 5

**Summary:**

This paper aims to address how to improve the inference capabilities of diffuse multimodal large language models (dMLLM). The core contribution is a novel training-free decoding strategy called ST-Veto, which leverages spatiotemporal lexical information to guide generation. Specifically, this method employs second-order Taylor expansion to predict confidence dynamics (temporal stability) and utilizes image attention quality to evaluate visual foundation (spatial reliability). During the unmasking process, lexical terms identified as unstable or with weak foundations are rejected and swapped with safer candidate terms.

**Compliance With Llm Reviewing Policy:**

Affirmed.

**Final Justification:**

rebuttal have solved my question

**Key Questions For Authors:**

(1) Why choose a second-order Taylor expansion? Have you observed diminishing returns or instability when using higher-order expansions? How does it differ quantitatively from simple first-order difference (delta) rejection?

(2) How does the method perform when the generated length L varies greatly? For very long sequences, is it necessary to adjust the rejection ratio to prevent overcorrection?

(3) If possible, add more comparative experiments and datasets.

If the author can address these concerns of mine, I will raise the score.

**Limitations:**

(1) dllm and ar have similar problems, namely text preference and neglect of image. The paper aggregates the attention of tokens in the middle layer of Transformer to image tokens, which may cause the model's recall to decrease. Many previous AR works have such defects, such as Middle[1], PAI[2], EAH[3]

- [1]Devils in middle layers of large vision-language models. CVPR 2025

- [2]Paying more attention to image. ECCV 2024

- [3]Seeing clearly by layer two. EMNLP 2025

(2) The motivation for choosing the second-order Taylor expansion is unclear. Why choose the local four-step?

(3) Since it is to improve the reasoning ability, the dataset should be universal. For example, MME comprehensive evaluation should be added, as well as illusion evaluations such as CHAIR and POPE. In addition, the comparison method should not be limited to the cot method. Some decoder training-free comparisons should be added, such as EAH[3], OPERA[4], Farsight[5] which also alleviate text bias.

- [4]OPERA: Alleviating Hallucination in Multi-Modal Large Language Models via Over-Trust Penalty and Retrospection-Allocation
- [5]Seeing Far and Clearly: Mitigating Hallucinations in MLLMs with Attention Causal Decoding

**Strengths And Weaknesses:**

(1) This paper correctly points out that AR-style inference methods are structurally mismatched with diffusion decoding. The proposed ST-Veto method addresses the unique property of dMLLM (global label visibility across steps).

(2) It innovatively uses Taylor expansion to model confidence evolution. Leveraging the history of confidence scores to detect instability (via residuals) provides a plausible method for identifying potential mislabels without additional training.

(3) This training-free approach makes it easy to deploy. Unlike multi-stage AR adaptive methods, its inference speed is comparable to baseline decoding.

---

> ### Author Rebuttal · Authors · 2026-03-31
>
> Thank you for your insightful review. We have carefully considered your comments and appreciate the opportunity for this discussion. Below is our rebuttal to the "Key Questions", and we would be grateful if you could review it once more.
>
> **Key Questions**
>
> 1.First, we thank you for highlighting the key contributions of our work. Our main idea is to leverage the ability to observe and modulate the generation process at each timestep t in diffusion LLMs, in contrast to AR LLMs. In determining whether to use first-order differences (delta) or higher-order Taylor expansions, we found that the optimal choice varies slightly across models and benchmarks. However, overall, the second-order Taylor expansion provided the most appropriate balance. The experimental results are shown below.
>
> Due to space limitations, and in accordance with the ICML rebuttal guidelines, we kindly ask for your understanding that we refer you to our responses to other reviewers (MBf7, rb9X) for more detailed analysis.
>
> 2.This is a critical point to consider. The optimal L may vary slightly depending on how the hyperparameter (max_new_tokens) was set during training for d(M)LLMs. The experimental results when L=512 are as follows, using the same experimental setup.
>
> | Dataset   | L/T     | Base | ST-Veto |
> |-----------|---------|----------|---------|
> | mmbench   | 512/256 | 32.54%   | **38.12%** |
> | scienceqa |         | 41.22%   | **48.72%** |
> | m3cot     |         | 30.58%   | **31.76%** |
> | vstar     |         | 36.65%   | **42.20%** |
>
> Furthermore, as shown in Table 4, increasing 𝐿 does not raise the Veto & Swap ratio, indicating that overcorrection is not a concern. This stability comes from our Veto & Swap Policy, which avoids directly modifying attention or logits. Unlike prior approaches that are sensitive to hyperparameters and prone to overcorrection, our method improves robustness by only modifying token selection.
>
> 3.We appreciate your insightful feedback. We include additional baseline and benchmark results below. However, most existing methods are designed for autoregressive models, creating a mismatch with d(M)LLMs. As a result, the current performance gap may not fully reflect our method’s strengths. A more definitive comparison will be possible as decoding methods tailored to d(M)LLMs become more established.
>
> (1)Experiments on a new benchmark using the LaViDa model with a 128/64 configuration. It should be noted that this performance is based on an inference mode that differs from the results reported in previous papers, where max_new_tokens was set to 4 or 8 to extract only the final answer.
>
> Due to space limitations, and in accordance with the ICML rebuttal guidelines, we kindly ask for your understanding that we refer you to our responses to other reviewers (MBf7, rb9X) for more detailed analysis.
>
> (2)Performance comparison with new baselines.Although not originally designed for dMLLMs, EAH shows the strongest improvement among baselines, suggesting that enhancing image attention in shallow layers is well aligned with dMLLMs. In contrast, methods like OPERA (beam search) and Farsight (causal mask modification) are less compatible, resulting in performance close to the baseline.
>
> | Benchmarks   | L/T     | Base | EAH    | OPERA  | Farsight | ST-Veto |
> |-----------|---------|----------|--------|--------|----------|---------|
> | mmbench   | 256/128 | 31.55%   | 33.89% | 31.92% | 31.67%   | **38.25%** |
> |           | 128/64  | 33.96%   | 34.96% | 34.03% | 34.23%   | **38.05%** |
> | scienceqa | 256/128 | 40.06%   | 43.44% | 41.12% | 40.85%   | **48.54%** |
> |           | 128/64  | 45.81%   | 45.52% | 45.75% | 45.22%   | **47.80%** |
> | m3cot     | 256/128 | 29.55%   | 29.45% | 29.56% | 29.65%   | **31.87%** |
> |           | 128/64  | 27.96%   | 27.87% | 28.11% | 27.88%   | **30.21%** |
> | vstar     | 256/128 | 37.17%   | 39.27% | 35.60% | 36.65%   | **41.36%** |
> |           | 128/64  | 35.60%   | 37.17% | 36.65% | 35.60%   | **42.94%** |
>
> **Limitations**
>
> 1.We appreciate your insightful comments and agree with your perspective. Our ablation results (Table 3; Figure 3(c,d)) suggest that temporal information plays a more significant role than image attention. However, vetoing low-attention tokens and swapping them with high-attention ones still yields consistent improvements. We acknowledge that visual grounding in d(M)LLMs requires further study and will include this in the limitations.
>
> 2.This is a key aspect of our study, also raised by Reviewer MBf7. Given that d(M)LLMs control generation via max_new_tokens and our setup uses T = 32 or 64, a 4-step history provides the best cost–performance trade-off. We acknowledge this was unclear and will include both the results and discussion in the revision.
>
> 3.We agree that limiting baselines to CoT methods is insufficient and that comparisons with training-free decoding methods are necessary. Please refer to our response to Question 3 in the Key Questions section for details.

---

> > ### Author Rebuttal · Reviewer_tKTA · 2026-04-02
> >
> > thanks for your rebuttal, please append the rebuttal content to the revision. I will raise my score.

---

> > > ### Author Response · Authors · 2026-04-05
> > >
> > > We sincerely thank you for taking the time to review our rebuttal positively and for increasing your score. Should you have any further comments or wish to continue the discussion, we would be happy to engage.
> > >
> > > Thank you.
> > >
> > > The Authors

---

### Official Review · Reviewer_J42k · 2026-03-01

**Soundness:** 2
**Presentation:** 3
**Significance:** 2
**Originality:** 2
**Overall Recommendation:** 3
**Confidence:** 3

**Summary:**

The paper proposes ST-Veto, a training-free decoding policy for diffusion MLLMs that vetoes unstable or weakly grounded tokens before unmasking, using a Taylor-based temporal residual and an attention-based visual grounding score. Vetoed tokens are swapped with safer alternatives near the top-k boundary to stabilize early anchors.

**Compliance With Llm Reviewing Policy:**

Affirmed.

**Final Justification:**

In my view, this paper requires a substantial number of additional experiments, as I pointed out in my rebuttal. Moreover, the paper does not sufficiently establish the connection between the motivation and the proposed methods, nor does it convincingly demonstrate their necessity. I will maintain my original score.

**Key Questions For Authors:**

Please refer to the Weaknesses for the detailed rebuttal questions, if the rebuttal resolves these concerns, I would be willing to revise my score.

**Limitations:**

The paper includes an Impact Statement, but it does not discuss the method’s limitations in much detail.

**Strengths And Weaknesses:**

Strengths:
1. The paper targets a clear issue in diffusion MLLMs, early token decisions can become unstable anchors, and AR-style CoT does not directly fit the parallel denoising process.
2. The reported overhead is small, and throughput stays close to the base model in the provided efficiency table.

Weaknesses:
1. The ``safe token'' selection near the top-k boundary is largely heuristic, I would like more analysis of what kinds of tokens are being swapped in.
2. The paper reports average gains but lacks breakdowns by question type or visual property. Since this is an optimization-oriented paper, I would expect more scenario-specific analyses to clarify where the method actually helps. This makes it hard to understand where ST-Veto helps most.
3. Visual grounding is approximated by attention mass, but the paper does not provide enough failure cases showing when attention is unreliable or diffuse, how often veto mistakes happen, and how they affect answers.
4. I am curious why the swap ratio is relatively small yet the gains can be noticeable, the paper does not fully explain how replacing only a small fraction of tokens leads to such improvements (e.g., an early-anchor effect vs benchmark sensitivity).
5. The main paper is almost entirely quantitative, I would recommend adding more qualitative analyses, either in the main paper or in the appendix.

Minor Weaknesses:
1. The AR setting is not protocol-aligned (two calls), which may bias both speed and quality comparisons, please report them separately.
2. The method depends on several hyperparameters, but the authors does not provide a systematic sensitivity study, making it unclear how robust the gains are across these settings.

---

> ### Author Rebuttal · Authors · 2026-03-31
>
> Thank you for your insightful review. We have carefully considered your comments and appreciate the opportunity for this discussion. Below is our rebuttal to the "Weaknesses", and we would be grateful if you could review it once more.
>
> **Major Weaknesses**
>
> 1.We agree with you on the importance of qualitative analysis. Prior work (ParallelBench: Understanding the Trade-offs of Parallel Decoding in Diffusion LLMs, ICLR 2026) shows that d(M)LLMs tend to favor probabilistically “obvious” tokens during unmasking. Our results confirm this: the vetoed top-k tokens are mostly functional words (e.g., articles, prepositions, conjunctions), while the swapped tokens are mainly content words (e.g., nouns, verbs, adjectives) with higher semantic stability. We appreciate this feedback and will include these findings in the revision.
>
> **Veto Tokens**
>
> | Part of Speech | Ratio | e.g |
> |----------------|-------|----------------|
> | Determiner (Det) | 28.4% | the, a, an |
> | Preposition (Prep) | 19.1% | of, in, at, to |
> | Conjunction (Conj) | 14.6% | and, or, but |
> | ... | ... | ... |
>
> **Swapped Tokens**
>
> | Part of Speech | Ratio | e.g |
> |----------------|-------|----------------|
> | Noun | 29.7% | dog, table, color |
> | Verb | 18.4% | holding, placed, sitting |
> | Preposition (Prep) | 14.2% | in, on, between |
> | ... | ... | ... |
>
> 2.Thank you for your insightful feedback. We provide task-level analysis on MVBench below, with a more comprehensive evaluation included in the Appendix of the final revision. Since ST-Veto does not directly modify attention or logits, it shows minimal side effects across categories. Notably, it achieves:
>
> (1) Strong gains in structured image-text understanding (+20%p),
> (2) Significant improvements in relational and comparative reasoning (e.g., physical_relation +14.9%, attribute_comparison +12.1%),
> (3) Consistent gains in recognition and classification tasks (e.g., action +12.1%, image_topic +12.9%, image_style +9.9%).
>
> | Category                          | N   | Base | ST-Veto | Diff. |
> |-----------------------------------|-----|----------|--------|------------|
> | structuralized_imagetext_understanding | 282 | 36.88%  | 57.09% | **+20.21%** |
> | physical_relation                 | 94  | 34.04%   | 48.94% | **+14.89%** |
> | image_topic                       | 140 | 31.43%   | 44.29% | **+12.86%** |
> | action_recognition                | 215 | 28.37%   | 40.47% | **+12.09%** |
> | attribute_comparison              | 141 | 32.62%   | 44.68% | **+12.06%** |
> | ...                               | ... | ...      | ...    | ...        |
> | celebrity_recognition             | 396 | 25.51%   | 27.02% | +1.52%     |
> | spatial_relationship              | 177 | 28.25%   | 28.81% | +0.56%     |
> | attribute_recognition             | 264 | 35.61%   | 35.23% | -0.38%     |
> | image_emotion                     | 200 | 40.50%   | 40.00% | -0.50%     |
> | identity_reasoning                | 176 | 26.14%   | 25.00% | -1.14%     |
>
> 3.We agree that clearer explanation of visual grounding is needed. To handle cases where visual or temporal signals are unreliable, we introduce a Veto & Swap Policy. If the grounding signal does not meet our criteria, no swap is performed. By applying swaps only when both signals are reliable, this approach minimizes failure cases and ensures stable performance across all settings.
>
> 4.We attribute these results to an early-anchor effect. As shown in Figure 3(a,b), prioritizing spatio-temporally stable tokens leads to higher-confidence selections over time. Compared to direct attention or logit manipulation, our Veto & Swap Policy is more robust. While direct injection was unstable and sensitive to hyperparameters, our method improves reliability by selectively swapping tokens, resulting in consistent performance gains.
>
> 5.We appreciate this insightful suggestion. An extended qualitative analysis addressing these points will be added to the appendix for further clarity.
>
> **Minor Weaknesses**
>
> 1.Thank you for your insightful feedback regarding the challenges in our study. As suggested, we will categorize and include those elements separately. However, we will further emphasize in the main text or the limitations section that AR-based methods do not function as intended within dMLLMs during a single inference, making a direct comparison difficult.
>
> 2.We sincerely appreciate your insightful comment. Following your suggestion, we conducted additional experiments summarized below. Due to space limitations, and in accordance with the ICML rebuttal guidelines, we kindly ask for your understanding that we refer you to our responses to other reviewers (J42k and MBf7) for more detailed analysis.
>
> **(1)Experiment on using different thresholds instead of MAD**
>
> **(2)Ablation study on individual layers**
>
> **(3)Experiments on History Length**
>
> **(4)Experiments on a new benchmark**
>
> **Limitations**
>
> We will include the points you raised in the Limitation section of the revised version. Thanks a lot again.

---

> > ### Author Rebuttal · Reviewer_J42k · 2026-04-03
> >
> > Thank you for the rebuttal, but several of my concerns are still not resolved.
> >
> > 1. Fundamentally, this work is a decoding heuristic, many existing benchmarks rely on LLM/VLM-judged evaluation that can be sensitive to factors such as output length or style rather than answer correctness itself. The paper does not provide direct evidence or sufficient supplementary analysis showing that the generated answers are genuinely better than those of the base model. The only qualitative example Fig.1 is overly simple and not convincing. From my experience, model-based evaluation can often be biased by superficial factors rather than true reasoning improvements.
> >
> >
> > 2. After carefully reading the rebuttal, I have an additional concern. On benchmarks such as ScienceQA, the reported scores in the paper are only around the 40, which suggests that the model still makes errors quite frequently. In such a setting, I wonder whether some of the observed gains could simply come from decoding perturbations rather than genuine reasoning improvements. That is, if the model is already unstable and often chooses the wrong option, then changing the decoding policy may sometimes alter the prediction path and, by chance, turn an originally incorrect choice into a correct one. If so, the reported improvement would not necessarily indicate that the model has developed better reasoning ability.
> >
> >
> > 3. I checked the benchmarks mentioned in the paper, the description of V* Bench somewhat unusual. It appears to be a spatio-temporal reasoning benchmark, rather than a high-resolution VQA benchmark as described in the paper. This inconsistency raises concerns about the clarity and correctness of the experimental setup.
> >
> >
> > The early-anchor explanation seems to be plausible, but it is still not isolated empirically. Higher confidence over time is consistent with this hypothesis, but it is not direct evidence that a small number of early swaps is the main reason for the observed gains. Step-specific ablations or case studies would be needed to support this claim. As I mentioned above, under the current experimental design, I find it difficult to be convinced.

---

> > > ### Author Response · Authors · 2026-04-03
> > >
> > > First, we sincerely thank you for carefully reviewing our rebuttal and for your insightful comments. As you pointed out, we acknowledge that there are still unresolved aspects. We would like to respond once again to the issues you raised, and we would greatly appreciate it if you could kindly review our response despite your busy schedule.
> > >
> > > 1,This is a very valid point. However, there may have been a misunderstanding due to our insufficient explanation, as we did not use LLM/VLM-judged evaluation. To address this concern, we exclusively used benchmarks that require **a clear final answer** after reasoning and **did not employ any benchmarks where LLMs/VLMs act as judges**.
> > >
> > > The mmbench benchmark is large-scale with many subtasks, while M3CoT, V∗, and ScienceQA focus on multimodal reasoning. Since we do not rely on LLM/VLM judges, there is no bias from differences in answer style.
> > >
> > > Additionally, for discussion MLLMs, the **“max_new_tokens”** parameter is fixed, and the LaViDa-Reason model and MMADA-Mix-CoT model we used are trained with a “think” prompt such as: **“You should first think about the reasoning process in the mind and then provide the user with the answer. The reasoning process is enclosed within <think> </think> tags, i.e., <think> reasoning process here </think> answer here.”** Therefore, all methods produce outputs in a consistent format. As a result, there is no incidental performance gain due to differences in output format compared to the base method. Upon verifying the outputs of all methods, we confirmed that they consistently generate responses up to the final answer.
> > >
> > > 2.We appreciate the reviewer's thoughtful concern. Indeed, rigorous verification of performance gains in a regime where the model's absolute accuracy is not yet high is essential. To this end, we conducted an instance-level flip analysis to verify that the observed improvements are not attributable to random decoding perturbation.
> > >
> > > **(1). Statistically Significant Asymmetric Flip Ratio**
> > >
> > > If the improvements were merely due to decoding perturbation, the number of wrong→right flips and right→wrong flips should be approximately equal (ratio ≈ 1.0, Z ≈ 0). However, our paired instance-level analysis reveals a strongly asymmetric pattern:
> > >
> > > | Benchmark | Base Acc. | Ours Acc. | Δ | Improved (W→R) | Regressed (R→W) | Flip Ratio | Z-score |
> > > | --- | --- | --- | --- | --- | --- | --- | --- |
> > > | MMBench | 31.55% | 38.25% | +6.70% | 518 | 228 | **2.27** | **10.62** |
> > > | ScienceQA | 40.06% | 48.53% | +8.47% | 410 | 239 | **1.72** | **6.71** |
> > >
> > > The Z-scores of 10.62 (p < 10⁻²⁵) on MMBench and 6.71 (p < 10⁻¹⁰) on ScienceQA strongly reject the null hypothesis that flips are random. A random decoding perturbation cannot account for such a consistent directional bias toward correct answers.
> > >
> > > **(2). Consistent Improvement Across Fine-Grained Categories**
> > >
> > > If the gains were driven by stochastic noise, we would expect roughly half of the categories to improve and the other half to degrade. Instead, we observe:
> > >
> > > MMBench: 17 out of 20 categories show positive net improvement (binomial test p < 0.001).
> > > ScienceQA: 8 out of 9 grade levels and all 3 subjects show positive net improvement.
> > > This degree of cross-category consistency is incompatible with the random perturbation hypothesis.
> > >
> > > **(3). Invalid Output Analysis Rules Out Format Artifacts**
> > >
> > > We also verified that on both MMBench and ScienceQA, base and ours produce zero invalid (unparseable) outputs, ruling out the possibility that the gains stem from cleaner output formatting rather than better reasoning. The improvements are entirely from the model selecting different—and more frequently correct—answer options, further confirming that the observed gains reflect genuinely improved token-level decisions during generation.
> > >
> > > However, we acknowledge that the validation of our assessment was not sufficiently emphasized in the manuscript. In the revised version, we will faithfully incorporate all rebuttal feedback, including discussions with other reviewers on the MME, POPE, and MMU benchmarks. We sincerely appreciate your insightful comments once again.
> > >
> > > 3.We sincerely apologize for the incorrect citation. There exists a benchmark with the same name, but the benchmark we used in our experiments is **(“V∗: Guided Visual Search as a Core Mechanism in Multimodal LLMs,” CVPR, 2024)**. It is indeed a **high-resolution VQA benchmark**. We appreciate your sharp observation, and we will ensure that the correct reference is included in the revised paper. Thank you.
> > >
> > > We would like to once again express our sincere gratitude for your strong interest in our work and for your thoughtful and in-depth review. We are very pleased to have had such a meaningful discussion, which has been extremely helpful in strengthening our paper.
> > >
> > > Thank you.
> > >
> > > The authors

---

### Official Review · Reviewer_rb9X · 2026-03-12

**Soundness:** 3
**Presentation:** 2
**Significance:** 3
**Originality:** 3
**Overall Recommendation:** 4
**Confidence:** 3

**Summary:**

The paper discovers that diffusion multimodal LLMs prefers tokens that are unstable or visually ungrounded, and introduces a runtime method that swaps those tokens for more stable, image‑grounded alternatives to boost reasoning accuracy.
It introduces Spatio‑Temporal Token Veto, a training‑free decoding strategy that jointly exploits second‑order Taylor predictions of token confidence dynamics and image‑attention mass to veto temporally unstable or poorly grounded tokens during generation. This paper shows, across multiple state‑of‑the‑art dMLLMs and reasoning benchmarks, that ST‑Veto consistently improves accuracy without extra training or inference cost.

**Compliance With Llm Reviewing Policy:**

Affirmed.

**Final Justification:**

The author's reply has addressed most of the reviewer's concerns. The score will be maintained as 4.

**Key Questions For Authors:**

No.

**Limitations:**

No. It would be better to include more disccusions on limitations.

**Strengths And Weaknesses:**

Strengths:
This work proposes a novel, principled decoding‑time technique that leverages the unique temporal visibility of diffusion models, achieving significant gains while remaining entirely training‑free.
This paper provides ablations and interpretability studies that illustrate how ST‑Veto steers generation toward higher‑confidence, visually grounded token trajectories.
This paper discussed why the proposed method/mechanism works in method and experiments.

Weaknesses:
The method presented in this paper is fairly novel, and I do not have many further questions. It would be valuable to discuss the failure cases and limitations in more depth. Additionally, the paper is missing several essential ablation studies, such as varying the history length and examining the robustness of the threshold and does not include evaluations on multimodal benchmarks like MMMU or MathVista. Although the paper references supplementary material, the corresponding content is absent. Finally, the article lacks sufficient qualitative examples, and the figure captions could be made more self‑contained and descriptive.

---

> ### Author Rebuttal · Authors · 2026-03-31
>
> We sincerely thank you for the insightful comments. We fully agree with the points raised, as these observations accurately identify the shortcomings in our current manuscript. We are committed to incorporating all of your suggestions into the final version of the paper. To address your concerns within the limited rebuttal timeframe, we have conducted the following additional experiments.
>
> **1) Experiments on History Length.**
>
> Our approach, which exploits the timestep characteristics of d(M)LLMs for temporal stability, consistently outperformed the baseline. When evaluating performance across history lengths (H) of 2, 3, 4, and 5, the approach using H=4 was most effective in most cases. While H=5 occasionally showed high performance, it incurs a higher computational cost by requiring third-order derivatives.
>
> | Dataset   | L/T     | Baseline | Delta (H=2) | 1st Taylor (H=3) | 2nd Taylor (H=4) | 3rd Taylor (H=5) |
> |-----------|---------|----------|-------------|------------------|------------------|------------------|
> | mmbench   | 256/128 | 31.55%   | 36.89%      | 37.98%           | **38.25%**       | 38.12%           |
> |           | 128/64  | 33.96%   | 36.98%      | 37.38%           | 38.05%           | **38.11%**       |
> | scienceqa | 256/128 | 40.06%   | 46.36%      | 47.60%           | **48.54%**       | 47.55%           |
> |           | 128/64  | 45.81%   | 46.69%      | 47.25%           | 47.80%           | **47.91%**       |
> | m3cot     | 256/128 | 29.55%   | 30.03%      | 30.50%           | **31.87%**       | 31.55%           |
> |           | 128/64  | 27.96%   | 28.69%      | 29.21%           | **30.21%**       | 30.18%           |
> | vstar     | 256/128 | 37.17%   | 39.27%      | 39.27%           | **41.36%**       | **41.36%**       |
> |           | 128/64  | 35.60%   | 37.17%      | 41.36%           | **42.94%**       | 39.27%           |
>
> **2.Experiments on a new benchmark using the LaViDa model with a 128/64 configuration.**
>
> It should be noted that this performance is based on an inference mode that differs from the results reported in previous papers, where max_new_tokens was set to 4 or 8 to extract only the final answer.
>
> | Dataset     | Base   | CCoT   | DDCoT  | ICoT   | ST-Veto |
> |-------------|--------|--------|--------|--------|---------|
> | MME Exist.  | 183.33 | 185.00 | 176.67 | 181.67 | **187.00** |
> | MME Count   | 133.33 | 126.67 | 137.22 | 136.33 | **143.33** |
> | MME Pos.    | 86.67  | 90.55  | 81.67  | 89.33  | **91.67**  |
> | MME Color   | 141.67 | 148.33 | 149.17 | 147.67 | **150.00** |
> | MME Total   | 545.00 | 550.55 | 544.73 | 545.00 | **570.00** |
> | POPE        | 84.45  | 84.95  | 85.10  | 84.95  | **87.65**  |
> | MMMU        | 30.44  | 31.13  | 30.54  | 31.22  | **33.41**  |
>
> **3.Experiment on using different thresholds instead of MAD.**
>
> When $\sigma_c$ was replaced with 0.5MAD, MAD, and 1.5MAD, 1.5MAD showed lower performance by forming an excessively high cutoff line, making token veto & swap difficult. On the other hand, 0.5MAD showed better performance than 1.5MAD but worse performance than MAD by allowing too many tokens under the Swap & Veto condition.
>
> | Dataset   | L/T     | Baseline | 0.5MAD  | MAD     | 1.5MAD  |
> |-----------|---------|----------|---------|---------|---------|
> | mmbench   | 256/128 | 31.55%   | 36.45%  | **38.25%** | 36.12%  |
> |           | 128/64  | 33.96%   | 36.62%  | **38.05%** | 37.24%  |
> | scienceqa | 256/128 | 40.06%   | 47.45%  | **48.54%** | 47.22%  |
> |           | 128/64  | 45.81%   | 47.12%  | **47.80%**  | **47.80%** |
> | m3cot     | 256/128 | 29.55%   | 31.22%  | **31.87%** | 31.22%  |
> |           | 128/64  | 27.96%   | 29.96%  | **30.21%** | 28.88%  |
> | vstar     | 256/128 | 37.17%   | 40.61%  | **41.36%** | 38.22%  |
> |           | 128/64  | 35.60%   | 39.27%  | **42.94%** | 41.36%  |
>
> We will include these comprehensive experimental results in the appendix. We appreciate your valuable feedback on the limitations of our current paper.

---

> > ### Author Rebuttal · Reviewer_rb9X · 2026-04-05
> >
> > Thanks for the author's reply which addressed most of the reviewer's concerns. The reviewer will keep the score as 4.

---

> > > ### Author Response · Authors · 2026-04-05
> > >
> > > We sincerely thank you for taking the time to review our rebuttal and for your positive evaluation. Please feel free to let us know if you have any further questions or points for discussion.
> > >
> > > Thank you.
> > >
> > > The Authors

---

### Official Review · Reviewer_MBf7 · 2026-03-13

**Soundness:** 4
**Presentation:** 3
**Significance:** 4
**Originality:** 4
**Overall Recommendation:** 5
**Confidence:** 3

**Summary:**

This paper addresses the reasoning limitations of diffusion Multimodal Large Language Models (dMLLMs), identifying that autoregressive Chain-of-Thought methods fail to transfer due to structural mismatches with the parallel decoding paradigm. The authors propose ST-Veto, a training-free decoding-time method that leverages the unique ability of dMLLMs to observe all tokens simultaneously. ST-Veto identifies temporally unstable tokens via second-order Taylor expansion of confidence dynamics and filters weakly grounded tokens using image attention mass, applying a veto-and-swap policy to replace risky tokens with safer alternatives. Extensive experiments across LaViDa and MMaDA demonstrate improvements of up to 9% on multimodal reasoning benchmarks without additional training or generation cost.

**Compliance With Llm Reviewing Policy:**

Affirmed.

**Key Questions For Authors:**

1. Is the fixed 4-step history window optimal across all diffusion timesteps? How does the reliability of stability prediction vary between early high-noise and late refinement stages?

2. Does averaging image attention across fixed layers capture functional differences between early visual feature layers and deep semantic layers? Would selective attention heads provide better grounding signals?

3. As generation length increases, early unmasked tokens naturally receive diluted attention; does the visual grounding signal remain effective, and how does ST-Veto handle attention dilution in long-context scenarios?

**Limitations:**

No. While acknowledging reliance on imperfect internal signals, the paper inadequately discusses robustness under non-smooth confidence trajectories, semantic misalignment in visual attention, and extension to multi-image or video inputs beyond single-image scenarios.

**Strengths And Weaknesses:**

**Strengths**

- The paper precisely identifies the structural incompatibility between autoregressive CoT methods and the diffusion generation paradigm, designing spatio-temporal dual-signal reasoning enhancement mechanisms tailored to dMLLMs' unique advantage of global token visibility, thereby filling a critical research gap in diffusion model reasoning enhancement.

- The ST-Veto method is elegantly designed and practical, combining temporal signals (Taylor residuals monitoring confidence stability) and spatial signals (image attention assessing visual grounding) with negligible computational overhead (approximately 1% inference time increase), requiring no training or model architecture modifications.

- Comprehensive experimental validation covers two representative dMLLMs (LaViDa and MMaDA) across multiple reasoning benchmarks (M3CoT, MMBench, ScienceQA, V*Bench), with thorough ablation studies and dynamic analyses revealing how the method guides generation toward high-confidence, visually grounded trajectories.

**Weaknesses**

- The fixed 4-step history window for Taylor expansion lacks theoretical justification and adaptive mechanisms for different diffusion timesteps or task complexities.

- Image attention quality calculation uses a fixed-layer averaging strategy without accounting for functional specialization differences across attention heads and layers, and handling of attention sink phenomena remains insufficiently clarified.

- The method is primarily evaluated on single-image-text generation scenarios, with limited discussion of scalability to complex scenarios such as multi-image inputs, video understanding, or ultra-long text generation.

---

> ### Author Rebuttal · Authors · 2026-03-31
>
> Thank you for your insightful review. We have carefully considered your comments and appreciate the opportunity for this discussion. Below is our rebuttal to the "Key Questions", and we would be grateful if you could review it once more.
>
> 1.We agree that the history window length (H) is a crucial component. The experimental results regarding H are presented below. By leveraging the timestep characteristics of d(M)LLMs, our strategy for pursuing temporal stability has proven effective, consistently outperforming the baseline in all cases. When comparing the performance across H = 2, 3, 4, and 5, H = 4 yielded the best results in most scenarios. While H = 5 occasionally showed better performance, it incurs a higher computational cost as it requires third-order derivatives.
>
> | Dataset   | L/T     | Baseline | Delta (H=2) | 1st Taylor (H=3) | 2nd Taylor (H=4) | 3rd Taylor (H=5) |
> |-----------|---------|----------|-------------|------------------|------------------|------------------|
> | mmbench   | 256/128 | 31.55%   | 36.89%      | 37.98%           | **38.25%**       | 38.12%           |
> |           | 128/64  | 33.96%   | 36.98%      | 37.38%           | 38.05%           | **38.11%**       |
> | scienceqa | 256/128 | 40.06%   | 46.36%      | 47.60%           | **48.54%**       | 47.55%           |
> |           | 128/64  | 45.81%   | 46.69%      | 47.25%           | 47.80%           | **47.91%**       |
> | m3cot     | 256/128 | 29.55%   | 30.03%      | 30.50%           | **31.87%**       | 31.55%           |
> |           | 128/64  | 27.96%   | 28.69%      | 29.21%           | **30.21%**       | 30.18%           |
> | vstar     | 256/128 | 37.17%   | 39.27%      | 39.27%           | **41.36%**       | **41.36%**       |
> |           | 128/64  | 35.60%   | 37.17%      | 41.36%           | **42.94%**       | 39.27%           |
>
> Additionally, we report the Veto and Swap ratios for early (high-noise) and late (refinement) stages. As noted, these ratios are higher in the early stage. However, this does not mean ST-Veto is negligible later—early unmasked tokens significantly influence subsequent decoding steps, as shown in Figures 3(a) and 3(b). We acknowledge our original explanation was unclear and will include both the table and this clarification in the final revision. Thank you for your insightful suggestions.
>
> | Time Step Ratio | 64 Steps | 128 Steps |
> |-----------------|----------|-----------|
> | 0–17%           | 18.3%    | 19.6%     |
> | 17–33%          | 38.3%    | 37.4%     |
> | 33–50%          | 33.7%    | 33.4%     |
> | 50–67%          | 8.6%     | 6.9%      |
> | 67–83%          | 5.4%     | 3.5%      |
> | 83–100%         | 1.8%     | 1.0%      |
>
> 2.Thank you for highlighting this important point. We selected the L/3, L/2, and 2L/3 layers to address this, and our results show that combining these stages outperforms using individual layers. While optimal choices may vary by model, different depths capture complementary information, and their integration yields more robust performance. We acknowledge this was not clearly explained and will include both the rationale and results in the final revision. Thank you for your valuable feedback.
>
> | Dataset   | L/T     | Baseline | L/3     | L/2     | 2L/3   | ALL     |
> |-----------|---------|----------|---------|---------|--------|---------|
> | mmbench   | 256/128 | 31.55%   | 34.21%  | 36.08%  | 35.74% | **38.25%** |
> |           | 128/64  | 33.96%   | 35.83%  | 37.12%  | 37.68% | **38.05%** |
> | scienceqa | 256/128 | 40.06%   | 44.52%  | 46.71%  | 47.85% | **48.54%** |
> |           | 128/64  | 45.81%   | 46.95%  | 47.40%  | 46.85% | **47.80%** |
> | m3cot     | 256/128 | 29.55%   | 30.67%  | 31.23%  | 30.85% | **31.87%** |
> |           | 128/64  | 27.96%   | 29.12%  | 29.85%  | **30.37%** | 30.21%  |
> | vstar     | 256/128 | 37.17%   | 39.27%  | 40.31%  | **41.88%** | 41.36%  |
> |           | 128/64  | 35.60%   | 38.74%  | 40.31%  | 41.36% | **42.94%** |
>
> 3.As noted by Reviewer tKTA, dMLLMs, like AR models, tend to prioritize text over visual input, as seen in Figure 3(c,d) where image attention declines over time. Our ST-Veto addresses this via the Low Visual Grounding Veto, filtering tokens with high textual confidence but weak visual support. As a result, ST-Veto maintains higher image attention across all steps (Figure 3(c,d)). These tokens act as “visual anchors,” guiding generation toward higher-confidence paths, as shown in Figure 3(a,b). We appreciate your insight into this key contribution.
>
> **About Limitations Sections**
>
> We will include the points you raised in the Limitation section of the revised version. Currently, dMLLMs do not yet demonstrate sufficient performance or scalability to handle multi-image or video inputs, so this will be addressed as future work.
>
> We sincerely thank you again for your positive review of our work, and we hope our rebuttal has addressed all of your concerns. Thank you.

---

> > ### Author Rebuttal · Reviewer_MBf7 · 2026-04-07
> >
> > Thank the authors for their great efforts in addressing all my concerns. Most of my concerns and questions have been resolved.

---

> > > ### Author Response · Authors · 2026-04-07
> > >
> > > We sincerely appreciate your positive evaluation of our research. Based on the rebuttal and discussion, we are committed to submitting a strong revised paper. It was both a pleasure and an honor to have this discussion with you.

---

### Decision · Program_Chairs · 2026-04-30

**Decision:**

Accept (regular)

**Comment:**

ST-Veto is a technically solid work in multimodal reasoning for dMLLMs. It introduces a novel, practical decoding-time enhancement that leverages global token visibility inherent to diffusion models. Experimental results are comprehensive, with statistical analyses and ablation studies supporting the claims. Limitations are minor relative to the impact of the approach, and rebuttal effectively addresses reviewer concerns. Overall, my recommendation is accept.